# Open Sciatic Nerve Decompression for Compartment Syndrome after Prolonged Lithotomy Position: A Case Report

**DOI:** 10.3390/medicina58101497

**Published:** 2022-10-21

**Authors:** Se-Hwan Lee, Hong-Pil Hwang, Sun-Jung Yoon

**Affiliations:** 1Department of Orthopedic Surgery, Jeonbuk National University Medical School, Jeonju 54907, Korea; 2Research Institute of Clinical Medicine, Jeonbuk National University, Jeonju 54907, Korea; 3Department of Surgery, Jeonbuk National University Medical School, Jeonju 54907, Korea

**Keywords:** case report, sciatic nerve, compressive neuropathy, lithotomy position, compartment syndrome

## Abstract

*Background:* Position-related compressive nerve injury is a frequently reported complication of the lithotomy position. In contrast, compartment syndrome-induced neuropathy after lithotomy with prolonged surgery is rare and prone to misdiagnosis. This case describes the successful open decompression of sciatic neuropathy due to compartment syndrome after a prolonged lithotomy position. *Case presentation:* A 56-year-old male patient complained of an abnormal sensation in the lower leg and difficulty in dorsiflexion and plantarflexion of the left foot and toes after laparoscopic anterior hepatic sectionectomy for 16 h in a lithotomy position. Physical examination revealed severe pain and paresthesia below the distal left thigh. In manual muscle test grading, dorsiflexion and plantarflexion of the left ankle and toes were classified as grade 1. Computed tomography and magnetic resonance imaging showed ischemic changes in the mid-thigh posterior muscles, and the sciatic nerve was severely swollen at the distal thigh, which was compressed by the proximal edge of the well-leg holder. After debridement of the necrotic tissue and decompression of the sciatic nerve, the pain subsided immediately, and the ankle and toe dorsiflexion motor function improved to grade 4. *Conclusions:* Most case reports of compressive neuropathy associated with the lithotomy position have been related to conservative treatment. However, if a lesion compressing the nerve is confirmed in an imaging study and the correlation with the patient’s symptoms is evident, early surgical intervention can be an effective treatment method to minimize neurological deficits.

## 1. Introduction

Compressive nerve injuries can be a complication associated with lithotomy position [1]. However, compartment syndrome-induced neuropathy after lithotomy with prolonged surgery is rare and prone to misdiagnosis. Compartment syndrome of the lower leg can be a rare but severe complication of surgery performed with the patient in the lithotomy position [2]. Compartment syndrome can be misdiagnosed as a perioperative neural injury, which appears as a neurological deficit. Delays in diagnosis and proper treatment can lead to irreversible functional loss and life-threatening complications [3]. Here, we report a case of sciatic nerve compressive neuropathy due to lower leg compartment syndrome that occurred in a male patient aged 56 years after a 16-h laparoscopic surgery in the lithotomy position. This case describes successful open decompression of sciatic neuropathy due to compartment syndrome resulting from a prolonged lithotomy position.

## 2. Case Presentation

A 56-year-old male patient with hepatocellular carcinoma underwent laparoscopic anterior sectionectomy during follow-up hepatobiliary surgery in our hospital. The operation lasted approximately 16 h, and the patient proceeded to the lithotomy position with a well-leg holder supporting both the distal part of the thigh and popliteal fossa. There were no preoperative neurological deficits, and the procedure was completed without intraoperative complications. However, after the surgery, the patient complained of an abnormal sensation in the entire lower leg. A decrease in dorsiflexion and plantar flexion of the left foot was observed. On physical examination, the patient complained of paresthesia over the knee joint and below. In the manual muscle test (MMT) grading, dorsiflexion and plantarflexion of the ankle and toe were classified as grade 1.

Conservative treatment was immediately administered; the patient was treated with gabapentin and physiotherapy was performed under the suspicion of transient peroneal nerve palsy. However, the patient complained of paresthesia below the knee joint, and the leg holder may have compressed the posterior aspect of the thigh. Ultrasonography was performed on the posterior aspect of the thigh to rule out sciatic nerve compression. Moreover, there was no definitive evidence of sciatic nerve compression lesion. An ankle-stop brace was used to prevent steppage gait and equinus deformities. However, the patient complained of persistent pain and tenderness of the posterior thigh. On follow-up physical examination, a dark brown transverse line was observed where the leg holder had been compressing the mid-thigh, and a mass-like lesion was palpated below the line (Figure 1a). Moreover, there was no improvement in sensory and motor functions (Figure 1b).

Nerve conduction velocity/electromyography (EMG) was performed and showed a severe sciatic nerve injury at the distal thigh level. Moreover, on computed tomography (CT), ischemic changes were observed in the adductor magnus, biceps femoris long and short head, semimembranosus, and semitendinosus muscles 25 cm below the femoral head, and the length of the lesion was approximately 6 cm (Figure 2a,b). The sciatic nerve in the affected area was observed to be swollen compared to the contralateral side, and the intermuscular fat was accompanied by haziness. Ischemic changes were observed in the posterior thigh muscles on magnetic resonance imaging (MRI), similar to the previous CT scan. In addition, the sciatic nerve was severely swollen compared with the contralateral side, and the T2-weighted signal intensity was increased (Figure 3).

During outpatient follow-up, the patient’s foot drop showed no significant improvement, the sensory deficit was sustained, and severe tenderness with a visual analog scale (VAS) score of 7 was observed, along with a palpable mass in the posterior aspect of the left thigh. Despite conservative treatment, there was no improvement in pain or motor and sensory function. In addition, apparent muscle necrosis and sciatic neuropathy were observed in the imaging study; therefore, surgical intervention was planned.

Debridement and decompression for sciatic nerve surgery were performed nine weeks following the onset of symptoms. The patient was placed in the prone position, and a longitudinal skin incision of approximately 10 cm was made in the mid-portion of the thigh, 20 cm proximal to the knee joint. The posterior muscles of the thigh were exposed by retracting the subcutaneous layer and performing a fasciotomy. Intraoperative findings showed an accumulation of old hematoma and necrotic changes in the semitendinous, semimembranous, and biceps femoris muscles. Necrotic tissue was debrided with careful protection of the sciatic nerve. The sciatic nerve appeared to be severely compressed by the necrosis of the semimembranous muscle and fibrous tissue. Adhesiolysis was performed on the sciatic nerve, a biopsy was performed, and the decompressed sciatic nerve was released from the necrotic tissue. The sciatic nerve was severely compressed at the lesion site (Figure 4). Meticulous hemostasis was performed, and the semitendinous, semimembranous, and biceps femoris muscles were repaired using absorbable sutures 1–0. After saline irrigation, a closed suction drain (400 mL) was placed, the subcutaneous layer was repaired, and skin suturing was performed. Two days after surgery, passive range of motion exercises were started. The patient was discharged four days after surgery.

After surgery, the pain immediately decreased with a VAS score of 3, and the ankle and toe dorsiflexion MMT improved to a grade of 3. The sensation in the lower extremities was restored by more than 50% compared with that before the operation at two weeks. MMT performed six months after surgery showed improvement in ankle dorsiflexion grade 4 and toe dorsiflexion grade 4, and the VAS score decreased to 1 (Table 1).

## 3. Discussion and Conclusions

Lithotomy-related neurovascular complications of the lower limbs have been reported in association with nearly every surgery, including orthopedic, gynecological, urological, and laparoscopic colorectal procedures. Lithotomy-induced lower leg compartment syndrome is also called well-leg syndrome because the calf is under excessive pressure and held in a higher position than the heart during surgery by the well-leg holder [4]. In most cases, lithotomy position-induced neuropathy is caused by direct compression of the common peroneal nerve around the fibular head [1]. Therefore, as occurred in this case, sciatic neuropathy caused by compartment syndrome of the distal thigh is prone to misdiagnosis. In this case, common peroneal nerve palsy was suspected when ankle dorsiflexion MMT was measured as grade I immediately following prolonged lithotomy. However, an attempt to rule out sciatic neuropathy was made as the patient complained of paresthesia over the knee joint, and the leg holder may have also compressed the posterior aspect of the thigh. Ultrasonography was performed on the posterior aspect of the thigh to rule out sciatic nerve compression; however, there was no visible definitive evidence of sciatic neuropathy due to space-occupying lesions. Initially, the patient underwent conservative treatment under the assumption of a diagnosis of peroneal nerve palsy. However, the patient continued to complain of persistent pain and tenderness in the posterior thigh. Additionally, on physical examination scarring and suspected compression from the well-leg holder were observed in the same region of the posterior thigh. We subsequently performed further evaluation, including MRI and found a sciatic nerve compression lesion with necrotic muscle tissue. When a patient is in the lithotomy position for an extended period, the good-leg holder can compress the posterior compartment of the thigh at the distal level. Therefore, it is thought sufficient pads to prevent compression, and proper hip and knee flexion should be used in procedures using well-leg holders.

Lithotomy position-induced compartment syndrome is uncommon [2] but can occur when the tissue pressure within a muscle compartment exceeds the capillary perfusion pressure after excessive compression on the leg holder for an extended period [5]. This may lead to decreased perfusion and ischemic changes. When the ischemic condition persists, the tissues and muscles undergo necrosis, leading to fibrosis in those muscles and tissues. Tan et al. [6] found an initial increase in compartment pressure of over 18 mmHg with the placement of the leg within the leg holder. Additionally, Myers et al. [6] reported similar findings, showing an increase in compartment pressure when the limb was elevated from the supine to the lithotomy position. In addition, there is a 0.78 mmHg decrease in arterial pressure for every centimeter the limb is elevated above the heart [7]. Hypotension during surgery also decreases the end-arterial pressure and negatively affects perfusion pressure. In this case report, the patient had compartment syndrome at the location of the distal thigh lesion after a prolonged lithotomy position.

The symptoms of compartment syndrome may sometimes be ambiguous, and thus the condition can be diagnosed based on clinical and objective findings. The symptoms and signs of compartment syndrome are pain, pallor, paresthesia, paralysis, and pulselessness, also known as the 5 Ps. Pain is a prevalent early sign of acute compartment syndrome in awake and alert patients [8]. Compartment pressure measurements are helpful objective tools and are essential diagnostic adjuncts to act on clinical suspicion. However, these measurements must not be the first line in diagnosing compartment syndrome in alert patients with normal sensitivity. Considering differential diagnoses, including cellulitis, deep vein thrombosis, neuropraxia, and peripheral arterial injuries is crucial. EMG or biopsy can be performed in patients with advanced compartment syndrome. Observation of muscle necrosis or fibrosis helps narrow down the diagnosis and calls for a differential diagnosis of neuropraxia. As mentioned earlier, the patient in this case initially complained of left foot drop and lower-leg sensory changes, which may have been confused with peroneal neuropraxia. The lower leg had been externally rotated in the lithotomy position, allowing for compression of the peroneal nerve. Nevertheless, in this case, MRI showed muscle necrosis of the posterior muscle compartment of the thigh, and surgery revealed a sciatic nerve severely compressed by the necrosis of the semimembranous muscles at the distal thigh level.

The treatment of compartment syndrome aims to reduce the pressure in the compartment as soon as possible, with fasciotomy being the most widely used modality. If the tissue pressure increases by >30 mmHg, fasciotomy must be considered immediately and without delay. In this case, two to three days after surgery the necrotic tissue was removed again until the compartment. When there is no muscle necrosis, the skin can be loosely sutured using a secondary suture, or skin grafts can cover the wound with sufficient growth of the granular tissue. Some controversy exists over the treatment of fasciotomy for delayed diagnosis of compartment syndrome; however, some treatments are agreed upon [9]. The patient, in this case, underwent an open decompression of the sciatic nerve, with a delay of approximately nine weeks showing good functional recovery after the operation.

Most case reports of neuropathy after lithotomy have been related to conservative treatment. Recently, there have been reports of symptoms and functional improvement when alpha-lipoic acid R was administered to patients with carpal tunnel syndrome [10]. However, nerve compression in the lithotomy position is often caused by direct mechanical compression inadvertently induced by a positioner. In this case, the peroneal nerve is compressed between the positioning devices at the fibular head level, resulting in neurological deficits such as foot drop. However, depending on the anatomical location and cause of neuropathy, conservative treatment should not be maintained, if the symptoms do not improve or if surgical treatment is projected to be effective, direct decompression is required. Direct surgical decompression and neurolysis can improve neuropathy caused by muscle necrosis after prolonged lithotomy. Therefore, active treatment is required for functional recovery. Early surgical intervention may improve the prognosis of patients with neurological deficits caused by compressive neuropathy. In particular, surgery can be effective when a cause, such as pressure from a surgical positioner, is clear and when the pathological location confirmed in an imaging study correlates with symptoms.

Most of the literature regarding compressive neuropathy related to the lithotomy position describes conservative treatment. However, if a lesion compressing the nerve is confirmed in an imaging study and this correlates with the patient’s symptoms, early surgical intervention is an effective treatment method to minimize neurological deficits.

## Figures and Tables

**Figure 1 medicina-58-01497-f001:**
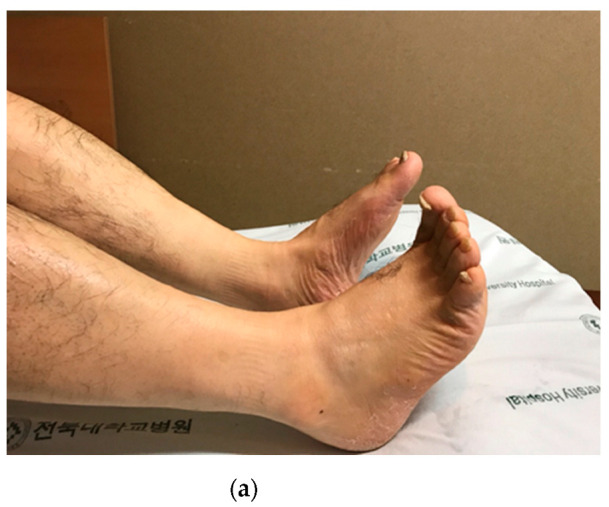
(**a**) 56-year-old male patient complained of abnormal sensation in the lower leg and difficulty in dorsiflexion and plantarflexion of the left ankle and toe. (**b**). Scarring (black arrowhead) due to prolonged compression by the well-leg holder is observed at the mid-thigh level, and a mass-like lesion is palpated below the line.

**Figure 2 medicina-58-01497-f002:**
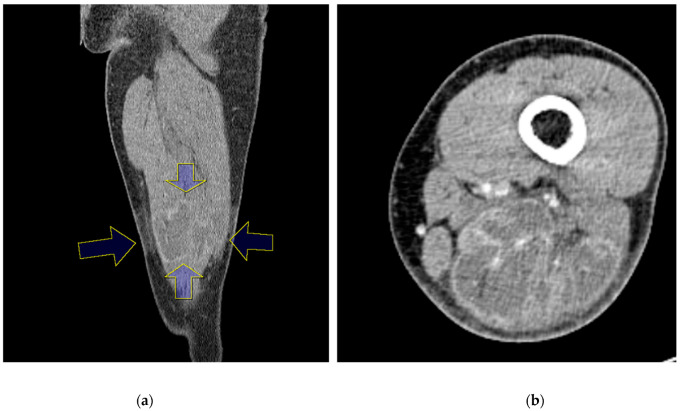
Computed tomography (CT) scan with contrast reveals ischemic changes in the adductor magnus, biceps femoris long and short head, semimembranosus, and semitendinosus muscles from 25 cm below the femoral head (blue arrow) in coronal view(**a**) and axial view(**b**).

**Figure 3 medicina-58-01497-f003:**
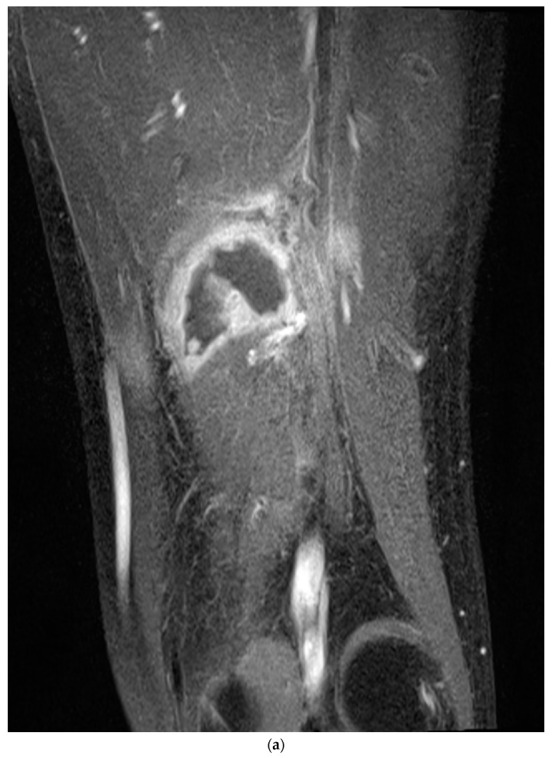
The T1-weighted magnetic resonance (MR) image (**a**) shows ischemic changes in the thigh posterior muscles, as in the previous CT scan. The sciatic nerve is severely swollen compared to the contralateral side, and the T2-weighted signal intensity (**b**) is increased (blue arrow). T1-weighted sagittal imaging (**c**) shows the length of the swollen sciatic nerve (green bracket).

**Figure 4 medicina-58-01497-f004:**
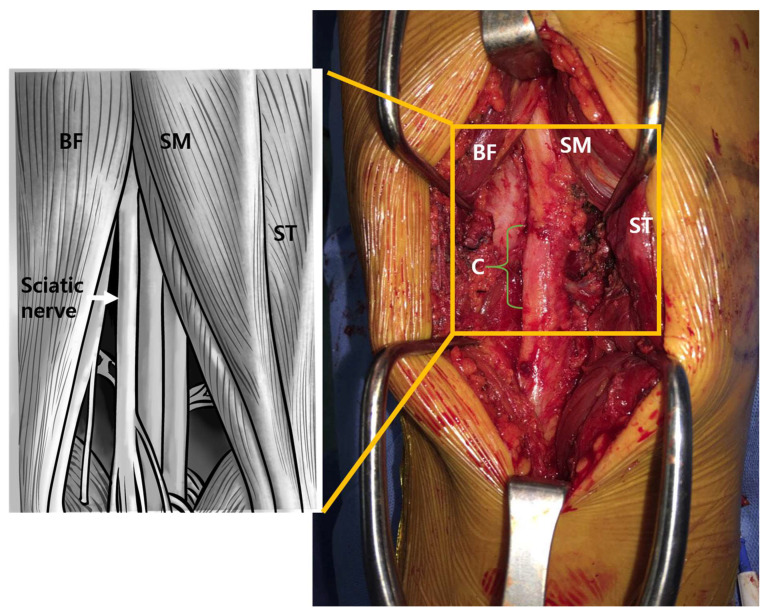
The intraoperative gross photo shows a severely compressed lesion of the sciatic nerve (green bracket). BF: Biceps femoris. SM: Semimembranosus. ST: Semitendinosus.

**Table 1 medicina-58-01497-t001:** Outcome and clinical features.

	Preoperative Scores	Immediate Postoperative Scores	Last F/U Scores
Knee extension	5	5	5
Ankle dorsiflexion	1	3	4
Toe dorsiflexion	1	3	4
Toe plantarflexion	1	3	4
Pain	VAS (7)	VAS (3)	VAS (1)

VAS: Visual analogue scale.

## Data Availability

Not applicable.

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
