# Peer review of "Open Sciatic Nerve Decompression for Compartment Syndrome after Prolonged Lithotomy Position: A Case Report"

_medicina, 2022, doi:10.3390/medicina58101497_

Round 1

Reviewer 1 Report

the article presents an interesting and difficult and equivocal case. The subject matter is interesting and it is useful to publish the treatment you have performed, given the success you have had. 

the article has been correctly developed throughout. I find it proper to specify that in the presence of a compressive neuropathy, especially if an ischaemic event is present as in your case, the use of neurotrophic and nutraceutical drugs can be useful, although, as you have shown, surgical decompression is the only possible solution. In recent literature, this concept has been stressed (PMID: 32296475) and should be mentioned in the article.

Author Response

Comment: the article presents an interesting and difficult and equivocal case. The subject matter is interesting and it is useful to publish the treatment you have performed, given the success you have had.

The article has been revised accordingly. I find it appropriate to specify that in the presence of compressive neuropathy, especially if an ischemic event is present, as in your case, the use of neurotrophic and nutraceutical drugs can be useful, although, as you have shown, surgical decompression is the only possible solution. In the recent literature, this concept has been stressed (PMID: 32296475) and should be mentioned in the article.

Response: Thank you for this comment. We have added a sentence to the discussion section as follows:

Recently, there have been reports of symptoms and functional improvement when alpha-lipoic acid R is administered to patients with carpal tunnel syndrome [11].

Reviewer 2 Report

This paper addresses important topic of position related complications after prolonged surgery. With no doubt surgery is treatment of choice in case of compartment syndrome. Therefore, my question to the authors is why during postoperative evaluation, the patient was suspected of common peroneal nerve palsy if on initial postoperative examination both dorsiflexion and plantarflexion were affected with MMT scores 1? Another issue is, whether ultrasound was performed as proximal at the thigh as 20 cm above knee joint if primarily common peroneal nerve palsy was suspected. US images from initial examination of the affected area should be included in the paper, while if no obvious signs at the level of injury was visible at US, this would be very interesting finding. This paper shows possible complications after prolonged positioning of the patient and will be interesting for the readers, however issues concerning initial diagnosis should be explained in detail prior to publication. Paper in this form suggest that US performed after 7 days of ischemic injury will not show any obvious signs of muscle damage and mass formation at the level of the injury.

Author Response

Response to the reviewers’ comments

Reviewer 1

Comment: the article presents an interesting and difficult and equivocal case. The subject matter is interesting and it is useful to publish the treatment you have performed, given the success you have had.

The article has been revised accordingly. I find it appropriate to specify that in the presence of compressive neuropathy, especially if an ischemic event is present, as in your case, the use of neurotrophic and nutraceutical drugs can be useful, although, as you have shown, surgical decompression is the only possible solution. In the recent literature, this concept has been stressed (PMID: 32296475) and should be mentioned in the article.

Response: Thank you for this comment. We have added a sentence to the discussion section as follows:

Recently, there have been reports of symptoms and functional improvement when alpha-lipoic acid R is administered to patients with carpal tunnel syndrome [11].

Reviewer 2

Comment:  This paper addresses important topic of position related complications after prolonged surgery. With no doubt surgery is treatment of choice in case of compartment syndrome. Therefore, my question to the authors is why during postoperative evaluation, the patient was suspected of common peroneal nerve palsy if on initial postoperative examination both dorsiflexion and plantarflexion were affected with MMT scores 1? Another issue is, whether ultrasound was performed as proximal at the thigh as 20 cm above knee joint if primarily common peroneal nerve palsy was suspected. US images from initial examination of the affected area should be included in the paper, while if no obvious signs at the level of injury was visible at US, this would be very interesting finding. This paper shows possible complications after prolonged positioning of the patient and will be interesting for the readers, however issues concerning initial diagnosis should be explained in detail prior to publication. Paper in this form suggest that US performed after 7 days of ischemic injury will not show any obvious signs of muscle damage and mass formation at the level of the injury.

Response: Thank you for this comment. Common peroneal nerve palsy was suspected when ankle dorsiflexion MMT was measured as grade I immediately after prolonged lithotomy. However, we attempted to rule out sciatic neuropathy because the patient complained of paresthesia over the knee joint, and the leg holder could have compressed the posterior aspect of the thigh. Moreover, skin lesions were observed on the posterior aspect of both thigh levels. Therefore, ultrasonography was performed on the posterior aspect of the thigh to rule out sciatic nerve neuropathy caused by space-occupying lesions. Unfortunately, there was no definitive evidence of sciatic nerve compression lesion. Initially, the patient underwent conservative treatment under the suspicion of peroneal nerve palsy. However, the patient complained of persistent pain and tenderness in the posterior thigh. In addition, on physical examination, scarring, suspected compression by the border of a well-leg holder was observed in the posterior thigh. Therefore, we performed further evaluation including MRI and found a sciatic nerve compression lesion by necrotic muscle tissue.

We have added the above description to the discussion section.
